# Differential Effects of Ammonium (NH_4_^+^) and Potassium (K^+^) Nutrition on Photoassimilate Partitioning and Growth of Tobacco Seedlings

**DOI:** 10.3390/plants11233295

**Published:** 2022-11-29

**Authors:** Oluwaseun Olayemi Aluko, Chuanzong Li, Guang Yuan, Tongjia Nong, Haiying Xiang, Qian Wang, Xuemei Li, Haobao Liu

**Affiliations:** 1Tobacco Research Institute, Chinese Academy of Agricultural Sciences, Qingdao 266101, China; 2Key Laboratory of Plant Stress Biology, School of Life Sciences, Henan University, 85 Minglun Street, Kaifeng 475001, China; 3Yunnan Academy of Tobacco Science, Kunming 650106, China

**Keywords:** NH_4_^+^-K^+^ concentrations, soluble sugar, starch, sugar-related enzymes, photoassimilate partitioning, tobacco seedlings

## Abstract

Plants utilize carbohydrates as the main energy source, but much focus has been on the impact of N and K on plant growth. Less is known about the combined impact of NH_4_^+^ and K^+^ nutrition on photoassimilate distribution among plant organs, and the resultant effect of such distribution on growth of tobacco seedlings, hence this study. Here, we investigated the synergetic effect of NH_4_^+^ and K^+^ nutrition on photoassimilate distribution, and their resultant effect on growth of tobacco seedlings. Soluble sugar and starch content peaks under moderate NH_4_^+^ and moderate K^+^ (2-2 mM), leading to improved plant growth, as evidenced by the increase in tobacco weight and root activity. Whereas, a drastic reduction in the above indicators was observed in plants under high NH_4_^+^ and low K^+^ (20-0.2 mM), due to low carbohydrate synthesis and poor photoassimilate distribution. A strong positive linear relationship also exists between carbohydrate (soluble sugar and starch) and the activities of these enzymes but not for invertase. Our findings demonstrated that NH_4_^+^ and K^+^-induced ion imbalance influences plant growth and is critical for photoassimilate distribution among organs of tobacco seedlings.

## 1. Introduction

Ammonium (NH_4_^+^), a major form of inorganic N, when applied in excess, leads to NH_4_^+^ toxicity, which culminates in decreased K^+^ uptake, leaf chlorosis, and stunted plant growth [1,2,3]). The adverse effects of NH_4_^+^ toxicity could be mitigated with an additional supply of K^+^ [4]. As such, balancing NH_4_^+^ and K^+^ becomes a prerequisite to improving plant growth. 

Another major factor that affects growth is the amount of assimilate synthesized and distributed among plant organs. Photoassimilate distribution is mainly influenced by several factors, including drought, temperature, and mineral nutrient deficit [5]. Of these nutrient elements, potassium (K^+^) [6] and nitrogen (N) [7] have been found to modulate photoassimilate partitioning between roots and shoots in a respective manner. Therefore, achieving a nutrient balance between K^+^ and NH_4_^+^ could be a tool for improving root-to-shoot biomass partitioning.

Root-to-shoot biomass partitioning is one of the inherent ways plants adapt to nutritional stress, which may subsequently influence plant growth [8,9]. Under optimal nutrient supply, root-to-shoot biomass partitioning is enhanced with resultant improvement in plant growth due to the diversion of a higher proportion of biomass to the leaves and stems, where sucrose is synthesised [8]. Although the significance of a balanced photoassimilate distribution to plant growth have been reported in several literature [10,11], less is known about the impact of K^+^ and NH_4_^+^-induced ion imbalance onroot-to-shoot biomass partitioning. 

Nutritional imbalances between K^+^ and NH_4_^+^ could negatively influence carbohydrate accumulation, distorting root-to-shoot biomass allocation. In N-deficient plants, more photosynthate (sucrose) is diverted towards the roots, resulting in an increasing root over shoot growth [12]. As such, instead of reserving the unloaded assimilates for growth, plant roots forage for more N in the soil or nutrient solution, as this could serve as the main adaptive strategy for root growth under such condition [13,14]. Plants subjected to low N supply also exhibit significantly higher leaf starch content and reduced plant growth due to the shortage in the supply of amino acid pool needed to sustain protein synthesis essential for the formation of new tissues [12]. On the contrary, excessive N-plants are mainly characterised by a wide range of NH_4_^+^ toxicity effect; low carbon retention in the root and higher retention in the shoot, reducing the sink (flowers and fruits) yield [15]. This adverse effect is ascribed to the flow of unassimilated NO_3_^−^ acropetally back to the shoot via xylem under excessive NH_4_^+^ condition. Thus, a balanced sucrose distribution between the source and sink (flowers or fruit) may act as a major yield-determinant factor. In contrast to the findings regarding lower root-to-shoot ratio, a higher root-to-shoot ratio was observed under excessive NH_4_^+^ in tobacco [16] and cucumber [17]. The aforementioned findings suggest that plant growth is disrupted when there is a shift in biomass partitioning between the source and sink, due to carbohydrate partitioning accrued from varying N application. 

With respect to K^+^, K^+^-deficient plants are characterised with reduced plant growth and photosynthetic rate associated with sucrose deposited in the source leaf [18] and a consequent distinct reduction in root sucrose and starch content. It is then deduced that changes in K^+^ status alters these physiological traits. In a comprehensive review on plant utilisation of sucrose, Aluko et al. [13] argued that changes in K^+^ concentration adversely affect the phloem-loading of sucrose, leading to a reduction in the root sucrose content. In all, N and K^+^ nutrition affects the photoassimilate export, and sugar accumulation in the leaves. 

The concentration of soluble sugars is controlled by sucrose metabolic system, which are critical to proper photoassimilate distribution among the plant tissues. The sucrose metabolic system is regulated by key enzymes: sucrose-phosphate synthase (SPS), sucrose synthase (SS), and invertase (Inv.) [19,20]. As the main photosynthetic product, sucrose is synthesised by sucrose-phosphate synthase (SPS), a key enzyme for carbohydrate partitioning. Sucrose synthase (SS) and invertase (Inv.) hydrolyses sucrose at the sink tissue (root, seeds, and younger leaves) [21], where the latter generate carbon and energy required for root (sink) growth. In addition, Inv. is involved in the distribution of sucrose within the sink organ. Therefore, Inv. is accessed as the master regulator of photoassimilate partitioning in response of plants to nutritional and environmental fluctuations [19]. 

Numerous studies have elucidated the influence of photoassimilate distribution on growth of plants exposed to drought [19,20], nitrogen [7], potassium [6], and other environmental cues. Moreover, literature has explored the impact of N and K on plant growth [3,22]; however, less is known about the combined impact of NH_4_^+^ and K^+^ nutrition on the photoassimilate distribution among plant organs, and the resultant effect of such distribution on growth of tobacco seedlings, hence this study. Here, we hypothesise that: (1) Photoassimilate alters the growth of tobacco seedlings under varying K^+^ and NH_4_^+^ (in a combined form) supplies, and (2) enzymes involved in sucrose conversion may be functionally associated with changes in photoassimilate and biomass partitioning. This present study investigated: (1) the synergetic effect of NH_4_^+^ and K^+^ nutrition on photoassimilate distribution, and their resultant effect on growth of tobacco seedlings, and (2) the relationship among biomass distribution, carbohydrate partitioning, and enzyme activity. 

## 2. Materials and Methods

### 2.1. Plant Materials and Growth Conditions

Tobacco seeds (*Nicotiana tabacum*) were sown in a potting soil mixture (soil/perlite, 3:1 *v*/*v*) under controlled climatic conditions (continuous light, temperature 24 °C). At the three-leaf stage, uniformly grown seedlings were transferred into hydroponic pots (48 cm × 22.5 cm × 3.5 cm) with 2 L of nutrient solution (one-fifth-strength Hoagland solution, 1/5 HS) for 6 days. The 1/5 HS, which was supplemented with 1 mM K^+^ (K_2_SO_4_ is the K^+^ source) had the following composition in mM: 0.35 MgSO_4,_ 0.2 NaH_2_PO_4_, 0.0125 H_3_BO_3_, 0.001 MnSO_4_, 0.0005 CuSO_4_, 0.001 ZnSO_4_, 0.0001 Na_2_MoO_4_, 0.01 Fe-EDTA, 1.4 Ca (NO_3_)_2_, and 0.15 CaCl_2_. NH_4_^+^ and K^+^ were supplied as (NH_4_)_2_SO_4_ and K_2_SO_4_, respectively.The experiment consisted of three levels of NH_4_^+^ (low, moderate, and high) combined with three levels of K^+^ (low, moderate, and high), making a total of nine NH_4_^+^ and K^+^ treatments, as follows: low NH_4_^+^ with low K^+^ (0.1-0.1 mM), moderate NH_4_^+^ with low K^+^ (2-0.1 mM), high NH_4_^+^ and low K^+^ (20-0.2 mM), lowNH_4_^+^ with moderate K^+^ (0.1-2 mM), moderate NH_4_^+^ with moderate K^+^ (2-2 mM), high NH_4_^+^ with moderate K^+^ (20-2 mM), low NH_4_^+^ with high K^+^ (0.2-10 mM), moderate NH_4_^+^ with high K^+^ (2-10 mM), and high NH_4_^+^ with high K^+^ (20-10 mM). We designed different levels of NH_4_^+^ and K^+^ supply based on the dose-response in our preliminary study. Plants were harvested after 15 days of NH_4_^+^-K^+^ treatments. All the treated groups were compared with each other. Moderate NH_4_^+^ and moderate K^+^ nutrition was also presented as the control for comparison with other NH_4_^+^ and K^+^ treated groups. In this study, a hyphen sign (“-”) was used in between NH_4_^+^ and K^+^ concentration to indicate the combined form of both (NH_4_^+^ and K^+^) treatments.

### 2.2. Sampling, Leaf Area, Root-to-Shoot Ratio, and Dry Weight Determination

At harvest, uniformly grown seedlings from each treatment were fractioned into (i) leaves, (ii) stems, and (iii) roots. Photographs of different plant parts were taken with a camera. Subsequently, the leaf area was determined using the ImageJ software (https://imagej.en.softonic.com/; accessed on 15 February 2022). Plant root was washed thoroughly once with 10 mM CaSO_4_ and twice in double-distilled water, and then the plant tissue sample was weighed. The dry weights of the measured samples were taken after oven-drying at 110 °C for 30 min and then 80 °C to a constant weight. The dry samples were crushed into fine powders with the mortar and pestle for K^+^ determination. Root to shoot ratio was calculated as root dry weight divided by the shoot dry weight (stem + leaf). The remaining part of the plant tissue (leaf, stem, and root) was collected, frozen, and stored in liquid nitrogen at −80 °C for enzymatic analysis.

### 2.3. Potassium and Ammonium Determination

Approximately 0.01 g of the grinded samples (leaves, stems, and roots) were weighed and digested in 8 mL of 0.5 M HCl for K^+^ concentration measurement. The suspension was homogenised at 25 °C, 100–150 rpm for 1 h, and filtered into a new centrifuge tube. The aliquot of the filtrate was used for K^+^ determination by flame photometry (6400 A). The readings obtained were used to calculate K^+^ concentrations in plant tissue, as follows:K^+^ (mmol g^−1^ DW) = ((A/M) × V × Dilution multiples × 0.001)/m
where
A = calculated concentration according to the readings on the standard curve (µg·mL^−1^);M = relative molecular mass of K^+^;V = reading volume (mL);m = dry weight (g).

For NH_4_^+^ concentration, the freshly harvested plant was partitioned into different plant parts (leaves, stems, and roots). The root was rinsed with 10 mM CaSO_4_ to eliminate any extracellular NH_4_^+^. Fresh plant tissue of ≤0.5 g was homogenised under liquid nitrogen, and 6 mL of 10 mM formic acid was added to extract NH_4_^+^. The suspension was allowed to sit for 5 min and then centrifuged at 4 °C and 12,000 rpm for 10 min. The supernatant was centrifuged repeatedly for about 3 times. The supernatant obtained from the last centrifugation step was diluted with 2.5 mL o-phthalaldehyde (OPA) solution, as previously described by Shi et al. [23]. The absorbance of the sample was measured at 410 nm using a spectrophotometer (UV-2550PC, AOE Instruments, Shimadzu Suzhou Instruments Mfg. Co, Ltd., Jiangsu, China). The reading obtained was used to calculate NH_4_^+^ concentrations in plant tissue as follows:NH_4_^+^ (µmol g^−1^ FW) = ((A/M) × V × Dilution multiples)/m
where
A = calculated concentration according to the readings on the standard curve (µg mL^−1^);M = relative molecular mass of NH_4_^+^;V = reading volume (mL);m = fresh weight (g).

### 2.4. Chlorophyll Content Measurement

After 15 days of NH_4_^+^-K^+^ treatment, chlorophyll content was measured according to the previous method [24]. The fourth leaf of each treatment was weighed (0.2 g) and incubated in 95% ethyl alcohol until the leaf strands became completely pale (approximately 48 h). The absorbance of the extract was measured at 665 nm and 649 nm using a spectrophotometer.

### 2.5. Oxidation-Reduction Potential Indicator of the Root

Using triphenyl tetrazolium chloride (TTC) method, root activity was measured, as previously described by Liu et al. [25] (with slight modifications. TTC method has been used to access the viability of metabolic active tissues, such as seeds’ and roots’ tissues [25,26]. The viability test relies on the reduction of water soluble TTC (with a standard oxidation potential of 80 mV) to an insoluble red 1,3,5-triphenyl formazan (TTF). This reduction could be ascribed to a TTC loss of electron, upon dehydrogenase activity in the root tissues. Thus, TTC, is used as a redox pigment for root activity measurement.

Approximately 0.5 g of the freshly weighed root was fully immersed in 5 mL of 0.4% TTC and phosphate buffer (adjusted to pH 7.0) and incubated at 37 °C for 3 h to accelerate the reduction of TTC to TTF. The resulting chemical reaction was halted by adding 2 mL of 1 mol L^−1^ sulphuric acid to each tube. Subsequently, the roots were removed from the tubes, gently patted with tissue paper, and then crushed with 3–4 mL ethyl acetate. The red supernatant and the root residue were moved into a new tube and made up to 10 mL of ethyl acetate. The absorbance was measured at 485 nm wavelength using a spectrometer (UV-2550PC, AOE Instruments, Shimadzu Suzhou Instruments Mfg. Co, Ltd., Jiangsu, China). For TTC standard curve, 0.2 mL of 0.4% TTC solution was added to a little amount of sodium sulphate (Na_2_SO_4_) powder, and mixed thoroughly to generate the redness. Then, 0.25 mL, 0.50 mL, 1.00 mL, 1.50 mL, 2.00 mL of this solution was discretely pipetted into a 10 mL volumetric flask, and the volume was made up with ethyl acetate to generate a standard colorimetric series containing 25 μg, 50 μg, 100 μg, 150 μg, and 200 μg, respectively. The absorbance is measured at a wavelength of 485 nm with a blank as a reference.
Tetrazole reduction strength (mg/g (fresh root weight)/h) = tetrazole reduction amount (mg)/[root weight (g) × time (h)]

The OD values were expressed as mg/g (fresh root weight)/h. 

### 2.6. Soluble Sugar and Starch Contents’ Determination

Sample extraction was performed and modified for soluble sugars and starch content determination according to Du et al. [20]. Approximately 0.02 g of ground leaf, stem, and root samples were homogenised with 80% (*v*/*v*) ethanol at 85 °C for 30 min. and centrifuged at 10,000× *g* for 10 min. The precipitates were extracted two to three times using 80% ethanol. The supernatants were combined and made up to 25 mL with 80% ethanol. The soluble sugar content was determined spectrophotometrically at A_620_ nm wavelength. The remaining ethanol-insoluble precipitates were used for starch extraction, as described by Kuai et al. [27]. The ethanol was removed, and the samples were diluted with 2 mL of distilled water, then incubated at 100 °C for 15 min. After cooling, 2 mL of 9.2 M was used to hydrolyse the leaf starch, and then centrifuged at 4000 rpm for 10 min. The pellet in the centrifuged solution was extracted again by adding 2 mL 4.6 M HClO_4_ to each of the samples. Thereafter, the supernatant was combined and made up to 25 mL volume of distilled water. The starch content was determined spectrophotometrically at a A_620_ nm wavelength using an anthrone reagent. Soluble sugar and starch concentrations of the leaves, stems, and root tissues were calculated and expressed in terms of mgg^−1^ DW. The proportion of sugar or starch in roots, leaves, and stems was calculated and expressed in a percentage.

### 2.7. Enzyme Extraction and Analysis

Plant samples were stored at −80 °C. Fresh plant tissue of ≤0.5 g was ground in a mortar with liquid nitrogen to analyse the sugar-related enzyme activity. The sugar-related enzymes (sucrose phosphatase synthase (SPS), sucrose synthase (SS), and acid invertase (Inv.)) were analysed using sucrose phosphorylase (SP) assay kit, Delphinose synthase (direction of synthesis; Ss-ii) kit, Soluble invertase (SAID/Vacuolar invertase) kit, and soluble acid invertase (S-AD)/Vacuolar invertase (G0517F) kit. Suzhou Greiss Biotechnology Co. Ltd., Suzhou, China. The manufacturer’s protocol was carefully followed.

### 2.8. Statistical Analysis

Data were analysed using the IBM SPSS Statistics 23 software. Variations among treatments were examined by one-way ANOVA using the LSD test at *p* < 0.05. Graphs and images were drawn using GraphPad Prism 6.0.

## 3. Results 

### 3.1. Effect of Different NH_4_^+^ and K^+^ Concentrations on the Dry Leaf, Stem, and Root Weight, and Root-to-Shoot Ratio (R:S) of Tobacco Seedlings

To investigate the effects of different NH_4_^+^-K^+^ concentrations on the growth of tobacco at the seedling stage, we measured the dry weight of the leaves, stems, roots, and root-to-shoot ratio (Table 1). The dry weight of leaf and root was improved when moderate NH_4_^+^ was combined with either moderate or high K^+^ (2-2 mM and 2-10 mM), respectively. Moreover, the dry weight of stem was enhanced under low NH_4_^+^ and moderate K^+^ (0.1-2 mM) nutrition, while other treatments maintained the stem weight without any adverse effect. Compared with the moderate NH_4_^+^-moderate K^+^ (2-2 mM) with increased biomass, the dry weight of leaves, stems and roots was significantly reduced when K^+^ was kept at a low level (0.1 mM and 0.2 mM) combined with NH_4_^+^ at low/moderate/high concentrations, respectively, and such a reduction was more pronounced in the stem. The decline in dry weight induced by high NH_4_^+^-low K^+^ in both the leaf and stem was alleviated in moderate or high K^+^ treated groups. Nevertheless, there were no observable changes in the root dry weight upon addition of moderate or high K^+^ when NH_4_^+^ was high. The dry weight of the whole plant was improved under moderate or high K^+^, especially when combined with the corresponding low or moderate NH_4_^+^concentration. However, a drastic reduction in the dry weight of the whole plant was observed when tobacco seedlings were subjected to low K^+^ stress. Although both dry shoot and root weight of tobacco seedlings were reduced when grown in K-deficient medium, yet, the root-to-shoot dry weight ratio was highest (15.58%) under such conditions due to the feedback effect of differential sensitivity of the plant organ to K^+^ nutrition.

### 3.2. Effect of Different NH_4_^+^ and K^+^ Concentrations on Chlorophyll Content, Leaf Area, and Root Activity of Tobacco Seedlings

Table 2 shows significant differences in the chlorophyll content, leaf area, and root activity of tobacco subjected to varying NH_4_^+^-K^+^ concentrations. A distinct increase in chlorophyll content was observed under high NH_4_^+^ at moderate (20-2 mM; 1.89 mg L^−1^) and high K^+^ (20-10 mM; 1.78 mg L^−1^) concentration, and are significantly different from other treatment groups. Further analyses demonstrated that the chlorophyll content was drastically reduced under low NH_4_^+^ with corresponding low K^+^ (NH_4_^+^-K^+^; 0.1-0.1 mM) and moderate K^+^ (NH_4_^+^/K^+^; 0.1-2 mM) concentrations, whereas other treatments, including the moderate NH_4_^+^ and moderate K^+^ (2-2 mM) had no significant effect on the chlorophyll content. Furthermore, leaf area was significantly increased under moderate NH_4_^+^ combined with moderate or high K^+^ (2-2 mM and 2-10 mM) concentrations compared to other treatments. Conversely, leaf area was markedly reduced in plants under low K^+^ concentrations, while other treatments, though statistically different, neither caused a marked increase or decrease in leaf area. Root activity was enhanced significantly under moderate NH_4_^+^ and moderate K^+^ nutrition, followed by low NH_4_^+^ and low K^+^ nutrition; however, root activity was drastically reduced under other treatments.

### 3.3. Influence of Different NH_4_^+^ and K^+^ Concentrations on Potassium Content (K^+^) in the Leaves, Stems, and Roots of Tobacco Seedlings

Significant differences in the K^+^ content of leaf, stem, and root of tobacco seedlings under varying NH_4_^+^ and K^+^ concentrations are presented in Table 3. Compared to the other treated group, K^+^ content in the leaf and stem peaks when supplied with moderate NH_4_^+^ and K^+^ (2-2 mM). In addition to the control, K^+^ content in the leaf and stem were also enhanced under high K^+^ with low and moderate NH_4_^+^ nutrition, respectively. As expected, K^+^ contents were lowest in the leaf, stem, and root of K^+^ deficient plants. The K^+^ content in the root peaks under low NH_4_^+^ and high K^+^ (0.2-10 mM) nutrition. Further analysis demonstrated that under high and moderate K^+^ with corresponding low or moderate NH_4_^+^ nutrition, the decrease in root K^+^ content was lower and significantly different from that observed under high NH_4_^+^ and low K^+^ nutrition.

### 3.4. Influence of Different NH_4_^+^ and K^+^ Concentrations on Ammonium Content (NH_4_^+^) in the Leaves, Stems, and Roots of Tobacco Seedlings

Significant differences in the NH_4_^+^ content of leaf, stem, and root of tobacco seedlings under varying NH_4_^+^ and K^+^ concentrations are presented in Table 4. The NH_4_^+^ content in leaf, stem, and root was markedly increased under high NH_4_^+^ with a corresponding low K^+^ supply; a further increase in K^+^ concentration (low to moderate to high) under such a high NH_4_^+^ resulted in a gradual reduction in the NH_4_^+^ content of the plant organs. For instance, in the roots, NH_4_^+^ content decreases with a corresponding increase in K^+^ concentration (from low (60.48 µmol g^−1^ FW) > moderate (54.28 µmol g^−1^ FW) > high (37.33 µmol g^−1^ FW). Although moderate NH_4_^+^ and moderate K^+^ had relatively higher NH_4_^+^ content compared to other treated groups, the hike in NH_4_^+^ content was the highest in NH_4_^+^-fed plants. 

### 3.5. Effects of Different NH_4_^+^ and K^+^ Concentrations on Soluble Sugar and Starch Content in the Leaf, Stems, and Roots of Tobacco Seedlings

To investigate the effects of varying NH_4_^+^ and K^+^ on the carbon distribution between the roots and shoots (leaf and stems), soluble sugar and starch contents of the leaves, stems and roots of tobacco seedlings were measured and are presented in Figure 1. There was a notable increase in soluble sugar content of the leaf under moderate NH_4_^+^ and moderate K^+^ (2-2 mM; 42.3 mg/g DW) nutrition, which is significantly different from other treatment groups. However, a significant reduction in the leaf soluble sugar content was observed under other treatments but to varying degrees. Compared with other treated groups, there was a significant decrease in the leaf soluble sugar content of high NH_4_^+^ (20 mM) with the corresponding K^+^ nutrition. A drastic decrease in leaf soluble sugar content was evident in plants under high NH_4_^+^ (20 mM) and corresponding high K^+^ (20-10 mM; 16.969 mg/g DW), moderate K^+^ (20-2 mM; 13.342 mg/g DW), and low K^+^ nutrition (20-0.2 mM; 10.377 mg/g DW). A similar trend was also observed in the stem and the root, although soluble sugar content was lower in the roots. The differential effects of NH_4_^+^ and K^+^ concentrations on the soluble sugar contents in the leaves and stems were the same as those of the roots.

The starch content of leaves, stems, and roots under different NH_4_^+^ and K^+^ levels are presented in Figure 2. Similar to the observations for soluble sugar content, the leaf starch content also peaks under moderate NH_4_^+^ and moderate K^+^ (2-2 mM; 2.515 mg/g DW); this is unlike the notable reduction in the leaf starch content observed under high NH_4_^+^ and low K^+^ nutrition (20-0.2 mM; 0.329 mg/g DW). Such reductions in starch content stemming from high NH_4_^+^ toxicity was alleviated in moderate or high K^+^ medium. For instance, a progressive increase in starch content of the leaves was evident in high NH_4_^+^ and moderate K^+^ (20-2 mM; 0.449 mg/g DW) or high NH_4_^+^ and high K^+^ (20-10 mM; 0.591 mg/g DW) medium. Although the differential effects of NH_4_^+^ and K^+^ concentrations on starch content in the leaves were the same as those in the stems and roots, the starch content is more accumulated in the leaves compared to the stems and roots. Furthermore, compared with moderate and high K^+^ nutrition, K^+^ deficient plants exhibited lower soluble sugar and starch content in the leaves, stems, and roots (Figure 1 and Figure 2).

### 3.6. Differential Effects of NH_4_^+^ and K^+^ Concentration on the Distribution of Soluble Sugar and Starch within the Plant Organs

The effects of NH_4_^+^ and K^+^ concentration on the proportion of soluble sugar and starch in various plant parts (leaf, stem, and root) were presented in Table 5 and Table 6, respectively. The differential effects of NH_4_^+^ and K^+^ concentration on the distribution of soluble sugar and starch within the plant organs are statistically different. The proportions of soluble sugars in both leaves and stems were significantly higher under low and moderate NH_4_^+^ combined with K^+^ at varying concentrations (low, moderate, high), respectively. A drastic reduction in soluble sugars diverted to the leaf and stem was observed under high NH_4_^+^, irrespective of the amount of K^+^ supplied. In all the treated groups, the amount of soluble sugars in the root was significantly lower than in the leaves and stem. The amount of root soluble sugar content was highest at high NH_4_^+^ and high K^+^ (20-10 mM: 26.735%) and lowest at moderate NH_4_^+^ and low K^+^ (20-0.2 mM: 15.774%). Moreover, soluble sugars were more diverted towards the roots of high NH_4_^+^ plants regardless of the K^+^ supply. In all, soluble sugar was more diverted towards the leaves than the stems and roots.

In the same lieu, the proportion of starch content peaks at moderate NH_4_^+^ and moderate K^+^, and lowest under high NH_4_^+^, regardless of the amount of K^+^ supplied: (48.967% versus 38.192%). Except for high NH_4_^+^, with reduced starch proportion in its leaves, the proportion of starch in every other treated concentration does not differ from each other (Table 6). In addition, the proportion of starch content in the stem peaks at moderate NH_4_^+^ and low K^+^, and lowest under high NH_4_^+^, regardless of the K^+^ supplied. However, no significant difference was found in other treatment groups. We deduced that irrespective of the concentration at which K^+^ was supplied, high NH_4_^+^ triggers a drastic reduction in starch content distribution in both the leaf and stem of tobacco seedlings. Contrarily, a marked increase in the proportion of root starch content was found in a high NH_4_^+^ medium, irrespective of the K^+^ supply.

### 3.7. Differential Effects of NH_4_^+^ and K^+^ Concentration on the Activities of Enzymes Related to Sucrose Synthesis and Degradation in Leaves and Roots

To better understand the differential effect of NH_4_^+^ and K^+^ concentration on carbohydrate utilisation and partitioning between leaves and roots, the activities of the three sucrose-related enzymes, sucrose phosphatase synthase (SPS), sucrose synthase (SS), and acid invertase (Inv.), were analysed and presented in Table 7. The activities of SPS and SS in both the leaves and root was highest under moderate NH_4_^+^ and K^+^ concentration and were significantly different from other treatment groups. Likely the enhanced SPS and SS activity in leaves and roots were also observed in plants under moderate NH_4_^+^ and high K^+^ nutrition. Further analysis of the leaves and roots showed significant reductions in the activities of SPS and SS when NH_4_^+^ was high, yet, a progressive increase in these enzyme activities was observed under high NH_4_^+^ with a corresponding increase in K^+^ (from low < moderate < high) concentrations. Regardless of the NH_4_^+^ supplied, the activities of SPS and SS were drastically reduced in K^+^- deficient leaves and roots compared with those of moderate and high K^+^ nutrition.

However, the invertase activities were lowest in low NH_4_^+^ and low K^+^, and moderate NH_4_^+^ and moderate K^+^ leaves and roots, respectively. Furthermore, invertase activities peaks in both leaves and roots of high NH_4_^+^ and low K^+^ plants, followed by high NH_4_^+^ and moderate K^+^ plants. Invertase activities in both root and leaves were increased in a stepwise manner when the plants were subjected to high NH_4_^+^ (20 mM) with a corresponding low K^+^ (0.2 mM) and moderate K^+^ (2 mM) nutrition.

### 3.8. Correlations between Sugar-Related Parameters and Ion Content (NH_4_^+^ and K^+^) among the Plant Organs 

The correlations between sugar-related parameters and ion content (NH_4_^+^ and K^+^) within the plant organs are presented in Figure 3. Leaf K^+^ content had a significant positive correlation with soluble sugar, starch, SPS, and SS of leaves and roots but was negatively correlated with the invertase activity in the root. However, only the soluble sugar and starch content in the root (*p* < 0.05), with sugar-related enzyme (except for invertase activity), demonstrated a positive correlation with K^+^ content in the stem and root. It is worth noting that the NH_4_^+^ content in the leaves, stems, and roots were negatively correlated with the soluble sugars (*p* < 0.01), starch, SPS activities in the leaves and roots, and SS activities in the root. Out of the sugar-related enzymes, only the invertase activity demonstrated a positive correlation (*p* < 0.01) with the NH_4_^+^ content.

## 4. Discussion

The optimal supply of nutrients such as NH_4_^+^ and K^+^ plays a crucial role in plant growth, evidenced by plant biomass. The plant biomass explains the degree of photoassimilate partitioning controlled by enzymes. Literature has explored the impact of N and K on plant growth [3,22]; however, less is known about the combined effect of NH_4_^+^ and K^+^ nutrition on photoassimilate distribution among plant organs, and the resultant effect of such distribution on growth of tobacco seedlings in a controlled environment. Therefore, we investigated the interactive effect of NH_4_^+^ and K^+^ on photoassimilate partitioning, since plants majorly rely on photoassimilate distribution for growth.

### 4.1. Dynamics of Dry Matter Partitioning under Varying NH_4_^+^ and K^+^ Nutrition

In the current study, the role of carbohydrate partitioning on growth of plant organs under varying concentrations of NH_4_^+^ and K^+^ nutrition has lent evidence again to tobacco plants during the seedling stage. Leaf and root biomass were significantly improved in the moderate NH_4_^+^ and moderate K^+^ (2-2 mM) medium; however, dry stem weight was only maintained under such conditions. Moreover, stem growth was significantly improved under low NH_4_^+^ (0.1 mM) and moderate K^+^ (2 mM) medium (Table 1), an indication that stems prefer low NH_4_^+^ relative to the leaf and root, which requires moderate NH_4_^+^ for growth. However, it has been previously reported that in young sugarcane subjected to varying NH_4_^+^ conditions, the shoot was more tolerant to high NH_4_^+^ than the root [28]. Thus, it could be inferred that the dry matter distribution in plants is dynamic and could be influenced by different stress factors such as NH_4_^+^ toxicity and the species of plants used [29]. The nutritional imbalance between NH_4_^+^ and K^+^_,_ especially in cases of high NH_4_^+^ and low K^+^ nutrition, results in impaired growth of plant organs. Our findings demonstrated that the toxicity effect of high NH_4_^+^ and low K^+^ on growth was more severe in the leaf and stem but was alleviated when supplied with moderate or high K^+^ (Table 1). This is in agreement with previous studies on *Arabidopsis* [23], rice [30], and barley [31]. Contrary to the expected notion that an extra supply of K^+^ would mitigate the toxicity effect of excess NH_4_^+^ on plant organ growth, the root growth of tobacco seedlings under high NH_4_^+^ nutrition was not improved by the additional supply of K^+^ in this study. This further suggests the tolerance capacity of the roots to NH_4_^+^. The reason for the halted root growth at high NH_4_^+/^moderate or high K^+^ could be part of a consequence of the energetic drain on root cells, catalysing the substantial futile cycling of both K^+^ and NH_4_^+^ under high nutrient supply [32,33]. 

Moreover, the resultant low root: shoot ratio due to the reduction in root weight and improved shoot (leaf and stem) under high NH_4_^+^ nutrition with an additional supply of K^+^ is in line with previous reports on wheat seedlings [4], maize [34], and sugarcane [28], which reported the same trend. The observed decrease in the root over shoot growth under high NH_4_^+^ nutrition could be because carbohydrates transported to the roots are used to assimilate ammonium and then relocated to the shoot in the form of amino acid or amide at the expense of the root [35]. In contrast, a higher root-to-shoot biomass ratio was observed in cucumber plants subjected to high NH_4_^+^ [17]. The variations in the results may be due to plant species. Furthermore, the dry weight of leaf, stem, and root of tobacco seedlings under K^+^ deficient nutrition was drastically reduced compared to that of seedlings under moderate or high K^+^ nutrition (Table 1); this agrees with previous studies [36,37]. In K^+^ deficient tobacco seedlings, treductions in dry weight were more pronounced in the stem compared to the root (Table 1), increasing root-to-shoot biomass ratio. The increased root-to-shoot ratio observed in K^+^-deficient plants would enhance its capacity to forage for nutrient richer patches, thereby improving the root growth at the expense of the shoot, and this supports the optimal biomass partitioning theory [8,38]. The dynamic changes in the root-to-shoot demonstrates the impact of NH_4_^+^ concentration on shoot and root growth in different plant species [4]. 

In addition, the combined impact of NH_4_^+^and K^+^ supply on the plant organs’ biomass, which in turn affects plant growth, may cause some alterations in physiological processes associated with plant growth and development. The inhibition of tobacco growth in response to NH_4_^+^ toxicity also led to a drastic reduction in growth variables including leaf area and root activity under high NH_4_^+^ and low K^+^ conditions; this is lined with findings on wheat [4]. Further analysis indicated that the increased chlorophyll content in high NH_4_^+^ medium was as a result of reduced leaf area, and this was affirmed by Walch-Liu et al. [16] and Koch et al. [39]. These changes in physiological parameters are linked with nutritional imbalance between NH_4_^+^ and K^+^, where the excess uptake of one causes an inhibition in the uptake of another, thus causing changes in the metabolic roles of these nutrients during plant growth and development. These findings provide enthralling evidence for growth response of plant organs to different concentrations of NH_4_^+^ and K^+^ nutrition due to their varying tolerance capacity and control of the physiological mechanisms involved, suggesting the application of the optimal nutrients for plants when grown in a controlled environment.

### 4.2. Varying NH_4_^+^ and K^+^ Tissue Content Impact on Growth of Tobacco Seedlings

The nutrient content of plant tissues is a key determinant of plant organs’ response to any form of stress. Plant tissue content of NH_4_^+^ and K^+^ varies under different nutritional regimes. Our study observed reductions in the K^+^ tissue content of tobacco seedlings under high NH_4_^+^ induced toxicity, in line with findings from Britto and Kronzucker [40] and Hoopen et al. [31]. The high NH_4_^+^-induced toxicity in plants triggered stunted growth and leaf chlorosis (dark greenish colouration), which were mitigated in plants with high K^+^ tissue content. This is in line with Hoopen et al. [31] and Szczerba et al. [2], who reported the same in rice, barley, and *Arabidopsis*. This is probably because potassium nutrients are actively involved in pathways underlying growth mechanisms, so low potassium nutrient uptake by plants may alter these mechanisms leading to suppressed growth. In the same line, reduced NH_4_^+^ tissue content was observed when the external supply of K^+^ concentration was high and vice-versa; this agrees with the findings of Hoopen et al. [31] in barley and *Arabidopsis*. In this study, regimes of NH_4_^+^ and K^+^ nutrition beyond the critical level (2-2 mM) culminates in K^+^- deficiency and excessive NH_4_^+^ conditions, and such nutritional imbalance leads to a drastic retarded plant growth. Meanwhile, this present study demonstrated that mitigating NH_4_^+^ toxicity in plants by striking a balance between NH_4_^+^-K^+^ (2-2 mM) improves crop growth and K^+^ content within the leaf, stem, and root. This is an indication that enhancing K^+^ tissue content improves the growth of tobacco seedlings. The aforementioned findings provide evidence once again on the usefulness on the application of optimal nutrients and nutrition balance for plant growth. Alterations in nutrient tissue content, root activity, leaf area and chlorophyll content due to varying concentrations of NH_4_^+^ and K^+^ nutrition may be linked with changes in photoassimilate partitioning fixed during photosynthesis.

### 4.3. Carbohydrate Partitioning Induced by Varying NH_4_^+^ and K^+^ Nutrition Affects Growth of Tobacco Seedlings

Photoassimilate partitioning and biomass allocation to different plant parts could be influenced by the supply of K^+^ or NH_4_^+^ nutrition [7,41,42]. Previous studies demonstrated that an adequate external nutrient supply of K^+^ [6] and NH_4_^+^ [43] induced an increase in soluble sugar and starch concentration in the leaves and roots. In addition, a marked increase in the sucrose concentration was observed in the K^+^ sufficient leaves [42,44]. In this present study, our findings revealed that soluble sugar and starch content peaks in tobacco seedlings under moderate NH_4_^+^ and moderate K^+^ (2-2 mM) concentration (Figure 1 and Figure 2). The increased soluble sugar and starch content distribution to the plant organs was conspicuous, as evidenced by improved biomass allocation within different plant organs (leaf, stem, and root). This was consistent with previous studies, which reported that a moderate supply of K^+^ [6] and NH_4_^+^ [7,45] enhanced soluble sugar and starch content, as well as its distribution within the plant organs, which in turn improved growth and plant biomass. The improved plant biomass at moderate NH_4_^+^ and K^+^ nutrition indicates an efficient carbohydrate distribution within the plant organs. Our current findings demonstrated that soluble sugar and starch concentration in tobacco plant organs are greatly influenced by deficiency or excess supply of NH_4_^+^ and K^+^ nutrition. The observed drastic reduction in soluble sugars and starch concentration of leaf, stem, and root in K^+^-deficient plants compared to that of moderate or high K^+^ plants could be attributable to reductions in sucrose synthesis [46]. Contrarily to our results, K^+^- deficient leaves exhibited increased soluble sugar concentration, restricting photoassimilate transport to the root and ultimately impedes growth [42,47]. The K^+^-deficiency could either trigger an increase or decrease in leaf sugar concentration [48]. The inconsistencies in leaf sugar concentration and its visible effects due to K^+^-deficiency may be attributable to plant species and the plant developmental stage. NH_4_^+^, per se, do not exert a direct negative effect on carbohydrate synthesis but, when supplied in excess, drastically reduces soluble sugar and starch concentration, reducing plant biomass [35]. 

The marked reduction in the soluble sugar and starch content of tobacco seedlings under high NH_4_^+^ nutrition could be strongly associated with the limited NH_4_^+^ assimilated under such conditions. However, this study did not consider the amount of NH_4_^+^ assimilated. Elevated NH_4_^+^ concentration in the shoot causes a marked reduction in the net carbohydrate assimilation of plants [49]. It is worthy to note that the additional supply of K^+^ to this high NH_4_^+^ medium slightly increased the soluble sugar and starch content of the plants, although not to an appreciable level. The observed increase in carbohydrates (as induced by K^+^) could be attributed to the mitigating effect of extra K^+^ on NH_4_^+^ toxicity, an indication that sufficient external K^+^ supply may be required for enhancing the plant-soluble sugar and starch concentration in tobacco seedlings during NH_4_^+^ toxicity occurrence. Taken together, we speculate that the reduction in plant growth due to a decrease in carbohydrate synthesis resulting from NH_4_^+^ toxicity is probably due to: less bioavailability of carbohydrates to be transported to the root, insufficient NH_4_^+^ assimilation in the root, and reallocation of the remaining free NH_4_^+^ to the shoot with a resultant reduction in carbohydrate synthesis. Carbohydrate and sugar synthesis are often regulated by SPS, SS, and Inv. enzymes.

### 4.4. Activities of Carbohydrate Biosynthesis Enzymes under Varying NH_4_^+^ and K^+^ Nutrition

Sucrose and starch biosynthesis and degradation are controlled by SPS, SS, and Inv. enzymes. The primary metabolic role of SPS is its involvement in sucrose biosynthesis, and the equilibrium constant of SPS activity supports the formation of sucrose phosphatase [19]. Moreover, SS activity in plants is crucial in sucrose storage and its utilisation for plant metabolic processes. SS activities help build new reservoirs for sucrose storage and the release of sucrose into cell wall polysaccharides for growth and respiratory process when needed [50]. The activities of these enzymes have been extensively studied under drought conditions and are often regulated by soluble sugar and starch concentration [20]. In a likewise manner, the activities of these enzymes were influenced by NH_4_^+^ and K^+^ nutritional regimes of varying concentrations in this study. Our findings revealed an increased activity of SPS and SS in plants under moderate NH_4_^+^ and moderate K^+^ medium could be due to the elevated soluble sugar and starch concentration in the leaves, stems, and roots.

Conversely, the activities of both enzymes (SPS and SS) were drastically reduced in plants under high NH_4_^+^ and low K^+^ nutrition; this reduction in enzyme activities may be due to decreased soluble sugar and starch concentration in the plants. These findings are suggestive that carbohydrate concentration (soluble sugar and starch) in plants is strongly associated with the activities of sugar-related enzymes. Interestingly, there was a progressive increase in activities of SPS and SS in plants under high NH_4_^+^ and moderate, or high K^+^ nutrition, and this could strongly be associated with the corresponding increase in external K^+^ concentration relative to that under low K^+^. Similar results were obtained with the study by Li et al. [51], which demonstrated an increasing nutrient supply of K^+^ enhanced the activities of SPS. Therefore, we could deduce that the improved plant K^+^ concentration (in leaf, stem and root) in high NH_4_^+^ and moderate or high K^+^ plants is the rationale behind the slight increase in enzymatic activities of SPS and SS (Table 7 and Figure 3). In this study, SPS activity was drastically reduced in low K^+^ or excessive NH_4_^+^ plants, indicating that high NH_4_^+^ and low K^+^ negate sucrose synthesis at the seedling stage. A similar trend has been reported in [7,36]. This suggests that sucrose biosynthesis due to activities of SPS is enhanced by moderate NH_4_^+^ and moderate K^+^ nutrient supply but are impaired in plants under high NH_4_^+^ and low K^+^ nutrition.

Moreover, there is evidence of increased SPS and SS activities in moderate K^+^ (K_2_O at 24 g m^−2^ soils) [6] and NH_4_^+^-plants [43]. In this study, moderate NH_4_^+^ and moderate K^+^ or high NH_4_^+^ and low K^+^ induce an increase or decrease in SS activity, respectively, which in turn reflects the capacity of roots to attract or inhibit photoassimilates correspondingly. It could therefore be deduced that SS activities are considered a good indicator of the ability of sink organs to attract photoassimilates. Invertase activity supports the distribution of carbohydrates to sink organs, regulates source-sink partitioning, and regulates plants’ response to environmental changes [52]. In this study, there was a marked increase in invertase activity of plants under high NH_4_^+^ supply (Table 7), which is in tandem with findings of Shen et al. [53]. This improvement could be linked with the fact that invertase is a key regulator of assimilation partitioning in plants’ response to environmental cues [54]. Reductions in the invertase activity may also be associated with reduced soluble sugar and starch content in the leaf, stem, and root of tobacco seedlings. The inverse relationship between invertase activity and carbohydrate content (soluble sugar and starch) (Figure 3) could be explained by the capacity of invertase to tolerate stressors despite its effect on carbohydrate partitioning. From the above findings, it could be deduced that alterations in plant growth and the metabolic pathways are strongly associated with the changes in photoassimilate partitioning fixed during photosynthesis, which are controlled by enzymes involved in carbohydrate synthesis. Photoassimilate partitioning, which plays a crucial role in plant growth and development are controlled by activities of SS and SPS in roots. The positive correlation between the enzymes and carbohydrate content in the plant organs (leaves, stems, and roots) lends evidence to the linear relationship observed in this study.

Carbohydrate is the major energy source required for plant growth; thus, enhanced soluble sugar and starch in moderate NH_4_^+^ and moderate K^+^ leaf, stem and root could be attributed to the improved plant growth at the early stage of development. Meanwhile, reduced and unbalanced carbohydrate distribution in high NH_4_^+^ (induced by NH_4_^+^ toxicity) and low K^+^ plants demonstrate the limitation in the energy required for growth under such conditions. This study highlights the critical need for optimal NH_4_^+^ and K^+^ concentration to facilitate plant growth via improved photoassimilate partitioning and activities of carbohydrate biosynthesis enzymes.

## Figures and Tables

**Figure 1 plants-11-03295-f001:**
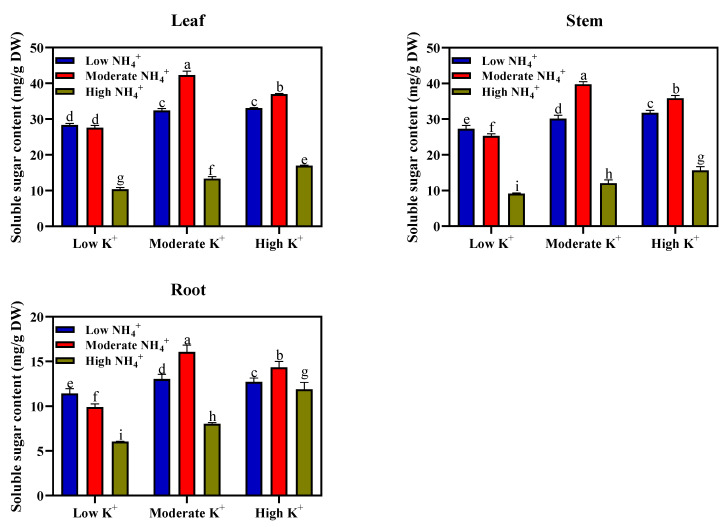
Soluble sugar concentration in leaves, stems, and roots (mg/g DW) of tobacco seedlings under different NH_4_^+^ and K^+^ levels. Different letters indicate significant differences between means ± SD at the *p* < 0.05 level (*n* = 9).

**Figure 2 plants-11-03295-f002:**
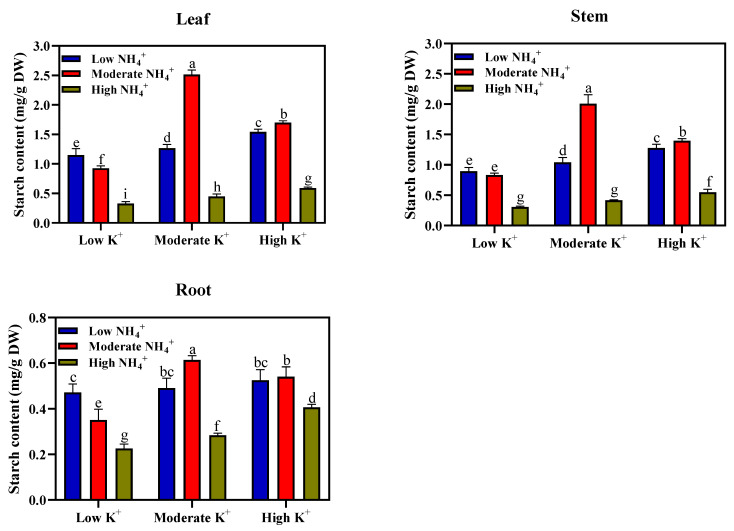
Starch contents of leaves, stems, and roots (mg/g DW) of tobacco seedlings under different NH_4_^+^ and K^+^ levels. Different letters indicate significant differences between means ± SD at the *p* < 0.05 level (*n* = 9).

**Figure 3 plants-11-03295-f003:**
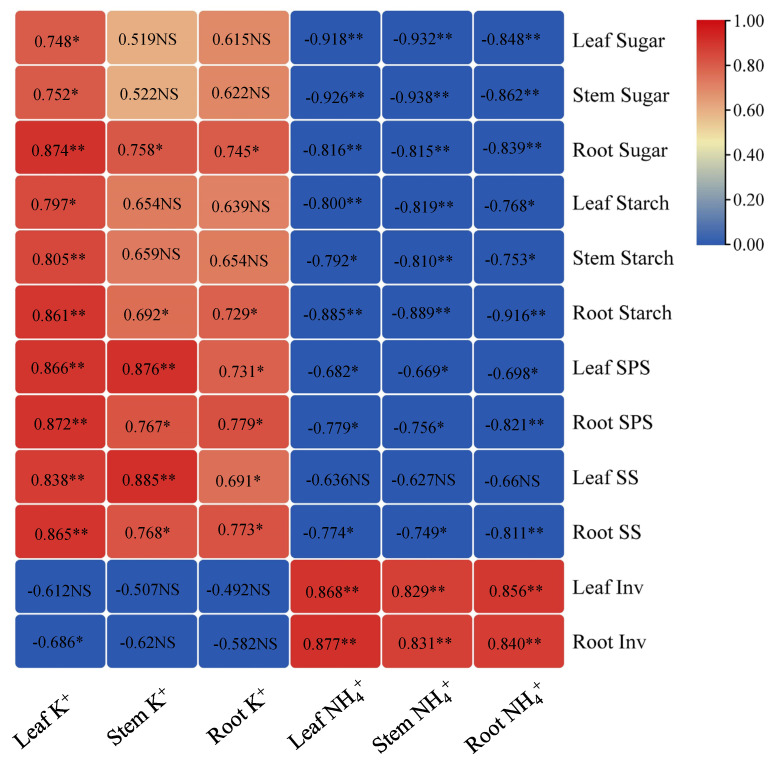
Relationship between sugar-related parameters and ion content (NH_4_^+^ and K^+^) of tobacco leaves, stems, and roots. The numbers in the figure represents the coefficient of determination, *R*^2^. The minus sign depicts a negative correlation between two indicators. * and ** demonstrate a positive relationship between the indicators and a significant difference at *p* < 0.05 and *p* < 0.01, respectively. “NS” represents not significant. The data employed in the correlation analysis represent the mean value under each treatment (*n* = 9).

**Table 1 plants-11-03295-t001:** Dry weights of leaves, stems, and roots, and root to shoot ratio (R/S) of tobacco seedlings under different NH_4_^+^ and K^+^ concentrations.

NH_4_^+^ Levels	K^+^ Levels	NH_4_^+^ (mM)-K^+^ (mM)	Leaf (g/Plant)	Stem (g/Plant)	Root (g/Plant)	Whole Plant (g/Plant)	R/S (%)
Low	low	0.1-0.1	0.24 ± 0.02 ^f^	0.03 ± 0.00 ^b^	0.04 ± 0.00 ^de^	0.30 ± 0.02 ^d^	15.58 ± 1.81 ^a^
Moderate	low	2-0.1	0.31 ± 0.02 ^e^	0.02 ± 0.00 ^b^	0.04 ± 0.00 ^d^	0.38 ± 0.02 ^d^	12.70 ± 0.71 ^b^
High	low	20-0.2	0.29 ± 0.02 ^e^	0.02 ± 0.00 ^b^	0.04 ± 0.00 ^f^	0.34 ± 0.02 ^d^	11.39 ± 1.05 ^b^
Low	Moderate	0.1-2	0.68 ± 0.02 ^c^	0.31 ± 0.27 ^a^	0.05 ± 0.00 ^c^	1.04 ± 0.26 ^b^	5.62 ± 1.20 ^e^
Moderate	Moderate	2-2	1.04 ± 0.07 ^a^	0.13 ± 0.00 ^b^	0.09 ± 0.01 ^a^	1.27 ± 0.06 ^a^	7.88 ± 1.12 ^cd^
High	Moderate	20-2	0.58 ± 0.02 ^d^	0.04 ± 0.00 ^b^	0.04 ± 0.00 ^d^	0.66 ± 0.02 ^c^	7.09 ± 0.34 ^de^
Low	High	0.2-10	0.77 ± 0.03 ^b^	0.10 ± 0.01 ^b^	0.08 ± 0.00 ^d^	0.94 ± 0.03 ^b^	8.75 ± 0.23 ^c^
Moderate	High	2-10	1.08 ± 0.04 ^a^	0.13 ± 0.00 ^b^	0.08 ± 0.00 ^b^	1.28 ± 0.03 ^a^	6.29 ± 0.37 ^e^
High	High	20-10	0.59 ± 0.04 ^d^	0.05 ± 0.00 ^b^	0.04 ± 0.00 ^ef^	0.68 ± 0.04 ^c^	5.75 ± 0.54 ^e^

Different letters indicate significant differences between means ± SD at the *p* < 0.05 level (*n* = 12).

**Table 2 plants-11-03295-t002:** Chlorophyll content (mg L^−1^), leaf area (cm^2^), and root activity mg/g (fresh root weight)/h of tobacco seedlings under different NH_4_^+^ and K^+^ concentrations.

NH_4_^+^ Levels	K^+^ Levels	NH_4_^+^ (mM)-K^+^ (mM)	Chl. Content	Leaf Area	Root Activity
Low	low	0.1-0.1	1.07 ± 0.02 ^f^	137.97 ± 5.01 ^f^	0.30 ± 0.00 ^b^
Moderate	low	2-0.1	1.46 ± 0.04 ^c^	105.57 + 3.14 ^g^	0.26 ± 0.00 ^c^
High	low	20-0.2	1.47 ± 0.02 ^c^	105.63 + 6.73 ^g^	0.11 ± 0.00 ^f^
Low	Moderate	0.1-2	1.17 ± 0.05 ^e^	237.21 + 0.62 ^c^	0.17 ± 0.01 ^d^
Moderate	Moderate	2-2	1.33 ± 0.02 ^d^	334.39 ± 1.14 ^b^	0.42 ± 0.01 ^a^
High	Moderate	20-2	1.89 ± 0.04 ^a^	183.81 + 3.32 ^e^	0.13 ± 0.00 ^e^
Low	High	0.2-10	1.32 ± 0.16 ^d^	211.50 + 4.83 ^d^	0.13 ± 0.01 ^e^
Moderate	High	2-10	1.48 ± 0.01 ^c^	354.78 + 4.23 ^a^	0.17 ± 0.00 ^d^
High	High	20-10	1.78 ± 0.03 ^b^	187.24 ± 5.87 ^e^	0.09 ± 0.00 ^g^

Different letters indicate significant differences between means ± SD at the *p* < 0.05 level (*n* = 12).

**Table 3 plants-11-03295-t003:** Potassium concentrations (mmol g^−1^ DW) in tobacco seedlings’ leaves, stems, and roots.

NH_4_^+^ Levels	K^+^ Levels	NH_4_^+^ (mM)-K^+^ (mM)	Leaf	Stem	Root
Low	low	0.1-0.1	0.66 ± 0.02 ^g^	0.75 ± 0.02 ^f^	0.38 ± 0.03 ^f^
Moderate	low	2-0.1	0.50 ± 0.01 ^h^	0.40 ± 0.02 ^g^	0.36 ± 0.03 ^f^
High	low	20-0.2	0.44 ± 0.01 ^i^	0.44 ± 0.03 ^g^	0.34 ± 0.03 ^f^
Low	Moderate	0.1-2	1.48 ± 0.04 ^d^	1.22 ± 0.07 ^e^	1.22 ± 0.10 ^bc^
Moderate	Moderate	2-2	1.77 ± 0.03 ^a^	2.01 ± 0.05 ^a^	1.19 ± 0.10 ^c^
High	Moderate	20-2	0.77 ± 0.03 ^f^	1.31 ± 0.05 ^d^	0.76 ± 0.02 ^e^
Low	High	0.2-10	1.63 ± 0.01 ^b^	1.23 ± 0.01 ^e^	1.41 ± 0.06 ^a^
Moderate	High	2-10	1.54 ± 0.02 ^e^	1.62 ± 0.07 ^b^	1.31 ± 0.05 ^b^
High	High	20-10	1.13 ± 0.03 ^e^	1.44 ± 0.04 ^c^	0.90 ± 0.09 ^d^

Different letters indicate significant differences between means ± SD at the *p* < 0.05 level (*n* = 12).

**Table 4 plants-11-03295-t004:** Ammonium (NH_4_^+^) (µmol g^−1^ FW) concentrations in tobacco seedlings’ leaves, stems, and roots under different NH_4_^+^ and K^+^ concentrations.

NH_4_^+^ Levels	K^+^ Levels	NH_4_^+^ (mM)-K^+^ (mM)	Leaf	Stem	Root
Low	Low	0.1-0.1	2.06 + 0.06 ^g^	1.06 + 0.06 ^h^	5.81 + 0.08 ^g^
Moderate	Low	2-0.1	18.59 + 0.24 ^d^	14.71 + 0.10 ^d^	45.15 + 0.12 ^c^
High	Low	20-0.2	51.67 + 0.14 ^a^	37.50 + 0.28 ^a^	60.48 + 1.00 ^a^
Low	Moderate	0.1-2	3.00 + 0.05 ^f^	0.86 + 0.06 ^h^	3.41 + 0.17 ^h^
Moderate	Moderate	2-2	4.81 + 0.22 ^e^	1.93 + 0.06 ^f^	10.75 + 0.31 ^e^
High	Moderate	20-2	37.21 + 0.35 ^b^	34.14 + 0.09 ^b^	54.28 + 0.81 ^b^
Low	High	0.2-10	1.57 + 0.04 ^h^	1.39 + 0.05 ^g^	1.92 + 0.03 ^i^
Moderate	High	2-10	3.22 + 0.40 ^f^	2.29 + 0.06 ^e^	7.40 + 0.21 ^f^
High	High	20-10	35.39 + 0.38 ^c^	28.68 + 0.09 ^c^	37.33 + 0.73 ^d^

Different letters indicate significant differences between means ± SD at the *p* < 0.05 level (*n* = 12).

**Table 5 plants-11-03295-t005:** Proportion of soluble sugar in leaves, stems, and roots (%) under different NH_4_^+^/and K^+^ concentrations.

NH_4_^+^ Levels	K^+^ Levels	NH_4_^+^ (mM)-K^+^ (mM)	Leaf	Stem	Root
Low	Low	0.1-0.1	42.30 ± 0.60 ^ab^	40.64 ± 1.37 ^a^	17.06 ± 0.84 ^c^
Moderate	Low	2-0.1	43.92 ± 0.41 ^a^	40.30 ± 0.73 ^a^	15.77 ± 0.32 ^d^
High	Low	20-0.2	40.58 ± 1.29 ^bc^	35.81 ± 0.75 ^b^	23.61 ± 0.60 ^b^
Low	Moderate	0.1-2	42.85 ± 0.86 ^a^	39.90 ± 0.63 ^a^	17.25 ± 0.35 ^c^
Moderate	Moderate	2-2	43.12 ± 0.89 ^a^	40.52 ± 0.52 ^a^	16.36 ± 0.54 ^cd^
High	Moderate	20-2	39.93 ± 2.144 ^cd^	36.02 ± 2.46 ^b^	24.05 ± 0.62 ^b^
Low	High	0.2-10	42.62 ± 0.36 ^ab^	40.95 ± 0.54 ^a^	16.43 ± 0.42 ^cd^
Moderate	High	2-10	42.43 ± 0.83 ^ab^	41.12 ± 0.31 ^a^	16.46 ± 0.53 ^cd^
High	High	20-10	38.18 ± 1.38 ^d^	35.09 ± 1.02 ^b^	26.74 ± 0.87 ^a^

Different letters indicate significant differences between means ± SD at the *p* < 0.05 level (*n* = 9).

**Table 6 plants-11-03295-t006:** Proportion of starch in leaves, stems, and roots (%) of tobacco seedlings under different NH_4_^+^ and K^+^ concentrations.

NH_4_^+^ Levels	K^+^ Levels	NH_4_^+^ (mM)-K^+^ (mM)	Leaf	Stem	Root
Low	Low	0.1-0.1	45.64 ± 1.98 ^ab^	35.58 ± 2.19 ^c^	18.78 ± 0.21 ^b^
Moderate	Low	2-0.1	43.94 ± 1.44 ^b^	39.45 ± 2.13 ^a^	16.62 ± 1.79 ^bcd^
High	Low	20-0.2	38.19 ± 3.30 ^c^	35.62 ± 0.88 ^c^	26.19 ± 2.47 ^a^
Low	Moderate	0.1-2	45.28 ± 1.78 ^b^	37.21 ± 1.52 ^abc^	17.51 ± 0.59 ^bc^
Moderate	Moderate	2-2	48.97 ± 1.56 ^a^	39.08 ± 1.72 ^ab^	11.96 ± 0.64 ^e^
High	Moderate	20-2	39.01 ± 2.51 ^c^	36.27 ± 1.47 ^bc^	24.72 ± 1.32 ^a^
Low	High	0.2-10	46.14 ± 1.01 ^ab^	38.18 ± 1.07 ^abc^	15.69 ± 0.76 ^cd^
Moderate	High	2-10	46.75 ± 1.18 ^ab^	38.40 ± 0.82 ^abc^	14.85 ± 1.00 ^d^
High	High	20-10	38.22 ± 0.57 ^c^	35.40 ± 1.98 ^c^	26.38 ± 1.43 ^a^

Different letters indicate significant differences between means ± SD at the *p* < 0.05 level (*n* = 9).

**Table 7 plants-11-03295-t007:** The activities of sucrose phosphatase synthase (SPS), sucrose synthase (SS), and acid invertase (Inv.) in leaves, stems, and roots subjected to different NH_4_^+^ and K^+^ concentrations.

NH_4_^+^ Level	K^+^ Levels	NH_4_^+^ (mM)-K^+^ (mM)	Leaf SPS	Root SPS	Leaf SS	Root SS	Leaf Inv.	Root Inv.
Low	Low	0.1-0.1	0.92 ± 0.02 ^e^	0.60 ± 0.01 ^e^	0.46 ± 0.01 ^d^	0.30 ± 0.02 ^d^	0.59 ± 0.01 ^f^	0.45 ± 0.02 ^ef^
Moderate	Low	2-0.1	0.75 ± 0.02 ^g^	0.49 ± 0.01 ^f^	0.34 ± 0.06 ^f^	0.25 ± 0.01 ^e^	0.80 ± 0.01 ^b^	0.57 ± 0.01 ^c^
High	Low	20-0.2	0.39 + 0.01 ^h^	0.19 + 0.01 ^h^	0.19 + 0.01 ^g^	0.09 + 0.00 ^g^	1.05 + 0.07 ^a^	0.83 + 0.03 ^a^
Low	Moderate	0.1-2	1.02 ± 0.02 ^d^	0.71 ± 0.02 ^d^	0.50 ± 0.01 ^cd^	0.36 ± 0.01 ^c^	0.68 ± 0.01 ^d^	0.51 ± 0.01 ^d^
Moderate	Moderate	2-2	1.71 ± 0.01 ^a^	0.85 ± 0.02 ^a^	0.90 ± 0.02 ^a^	0.42 ± 0.01 ^a^	0.62 ± 0.01 ^ef^	0.42 ± 0.01 ^g^
High	Moderate	20-2	0.78 ± 0.01 ^f^	0.42 ± 0.03 ^g^	0.39 ± 0.01 ^e^	0.22 ± 0.01 ^f^	0.83 ± 0.02 ^b^	0.61 ± 0.01 ^b^
Low	High	0.2-10	1.10 ± 0.02 ^c^	0.75 ± 0.01 ^bc^	0.52 ± 0.01 ^c^	0.37 ± 0.01 ^bc^	0.64 ± 0.01 ^de^	0.47 ± 0.02 ^e^
Moderate	High	2-10	1.30 ± 0.01 ^b^	0.75 ± 0.01 ^b^	0.64 ± 0.03 ^b^	0.38 ± 0.01 ^b^	0.74 ± 0.02 ^c^	0.45 ± 0.02 ^ef^
High	High	20-10	1.03 ± 0.04 ^d^	0.72 ± 0.02 ^cd^	0.51 ± 0.01 ^c^	0.36 ± 0.01 ^bc^	0.74 ± 0.03 ^c^	0.53 ± 0.02 ^d^

Different letters indicate significant differences between means ± SD at the *p* < 0.05 level (*n* = 9).

## Data Availability

The data presented here are available on request. Contact the corresponding authors for details.

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
