# Peer review of "Differential Effects of Ammonium (NH4+) and Potassium (K+) Nutrition on Photoassimilate Partitioning and Growth of Tobacco Seedlings"

_plants, 2022, doi:10.3390/plants11233295_

Round 1
Reviewer 1 Report
The manuscript by Aluko and collaborators reports an intensive study on the combined effect of NH4+ and K+ nutrition on photoassimilate distribution among plant organs, and the resultant effect of such distribution on growth of tobacco seedlings. Specifically, the authors hypothesized that (1) Photoassimilate alters the growth of tobacco seedlings under varying K+ and NH4+ (in a combined form) supplies, and (2) enzymes involved in sucrose conversion may be functionally associated with changes in photoassimilate and biomass partitioning. To assess these hypotheses, the authors grew young tobacco seedlings on different levels of NH4+ (low, moderate, high) combined with different levels of K+ (low, moderate, high) giving 9 nutritive treatments. After 15 d of growth, the authors measured several parameters on plants that were biomass production (leaves, stems, roots) and partitioning, chlorophyll, K+ and NH4+ contents, root activities and sugar-related enzymes. The experiment was well-designed and produced a huge amount of data which helps to understand the combined effects of NH4+ and K+ on plant physiology. The manuscript is very well written and I have only some minor comments which might help to improve the reading of the paper.
Minor comments
1. My first comment is about root activity assays (line 166, § 2.5).
What is measured with the assay ? An oxidative activity ? please give the general principle of measurement to precise this. In addition which solution is used to measure OD at 485 nm? The ethyl acetate without the root? What is TTF?
2. Line 186, you wrote “After cooling, 9.2 M and 4.6 M HClO4 were separately added to the samples” but which volumes? Please precise.
3. Figure 2: the y-axis could be the same for leaves and stems, as in Fig. 1.
4. Tables 5 and 6: please put only two digits after the unit.
5. The main paragraphs in the Discussion could have a subtitle giving the main idea which is discussed.
Reviewer 2 Report
The manuscript entitled “Differential effects of ammonium and potassium nutrition on growth and photoassimilate partitioning in tobacco seedlings” elaborates on the combined effect of ammonium and potassium nutrition on the growth of tobacco seedlings and highlights the need for optimal ammonium-to-potassium concentrations to facilitate the growth of tobacco seedlings via improved photoassimilate partitioning and balanced activities of carbohydrate biosynthesis enzymes.
Dynamics of dry weights, chlorophyll content, leaf area, potassium and ammonium concentrations, soluble sugar content, starch content and their proportions, vs sucrose phosphatase synthase, sucrose synthase and acid invertase activities in leaves, stems and roots of tobacco seedlings are presented and discussed under nine combinations of low, moderate, or high ammonium vs potassium levels. The reduced and unbalanced carbohydrate distribution under high ammonium concentrations vs low potassium ones provide barriers on the required energy for growth under the circumstances.
The manuscript is well written and mature for publication.
Author Response
Thank you for reviewing our manuscript and endorsing it for publication.
Reviewer 3 Report
This manuscript describes the synergetic effect of NH4+ and K+ nutrition on photoassimilate distribution, and their resultant effect on growth of tobacco seedlings. This study investigated the synergetic effect of NH4+ and K+ nutrition on photoassimilate distribution, and their resultant effect on growth of tobacco seedlings, and the relationship among biomass distribution, carbohydrate partitioning, and enzyme activity. Based on these results, the authors claim that NH4+ and K+-induced ion imbalance influences plant growth and is critical for photoassimilate distribution.
In my opinion, this manuscript merely describes the phenotypes of tobacco seedlings at varying levels of NH4+ and K+, making it difficult to find scientifically new points. In addition, the two hypotheses described in the introduction are not specifically mentioned in the discussion. If the scientific value of this manuscript must be stated, it would be to present the description of the phenomena obtained in this study in a more understandable way to the reader.
Since it is very difficult to understand the contents of Table1~6, the following revisions should be proposed.
--
1) Table 1~5 should be presented as relative values to NH4+ & K+ moderate values, and also as a heat map like the attached word document (alphabets of significant differences should also be included).
2) Table 1~6 of the initial draft should be presented as a Supplemental Table.
--

Round 2
Reviewer 3 Report
While this is my suggestion to make this paper more understandable, it would be better to present at least Table 8 as a heat map.
As an example, Figure 7 in the following paper would be helpful.
https://doi.org/10.3389/fpls.2022.1035254
